# Basic Methods of Cell Cycle Analysis

**DOI:** 10.3390/ijms24043674

**Published:** 2023-02-12

**Authors:** Anna Ligasová, Ivo Frydrych, Karel Koberna

**Affiliations:** Institute of Molecular and Translational Medicine, Faculty of Medicine and Dentistry, Palacký University in Olomouc, Hněvotínská 5, 779 00 Olomouc, Czech Republic

**Keywords:** cell cycle, labeled nucleosides, BrdU, EdU, markers of cell cycle phases, DNA labeling, time lapse microscopy

## Abstract

Cellular growth and the preparation of cells for division between two successive cell divisions is called the cell cycle. The cell cycle is divided into several phases; the length of these particular cell cycle phases is an important characteristic of cell life. The progression of cells through these phases is a highly orchestrated process governed by endogenous and exogenous factors. For the elucidation of the role of these factors, including pathological aspects, various methods have been developed. Among these methods, those focused on the analysis of the duration of distinct cell cycle phases play important role. The main aim of this review is to guide the readers through the basic methods of the determination of cell cycle phases and estimation of their length, with a focus on the effectiveness and reproducibility of the described methods.

## 1. Introduction

A common task of many research teams is the analysis of cell cycle progression through the distinct cell cycle phases. Various methods can be used for this purpose. While most of the high-throughput and relatively fast methods provide relative information about the mean values and no information or partial information about the variability of the measured parameters, some others provide both, albeit at the cost of a much lower analysis rate.

Information about cell cycle progression can be very useful, e.g., for understanding and predicting tumor development. Cancer cells frequently display unscheduled proliferation, genomic instability (with the increased DNA mutation rate and chromosomal aberrations), and chromosomal instability (changes in chromosome number) [1]. Dysregulation of the cell cycle is thought to be the first step in carcinogenesis, e.g., of renal cell carcinoma, important for tumor invasion and metastases [2]. Identification of changes in the progression through the cell cycle is also important during drug testing, as it can provide important data which helps to elucidate the mechanism of action of these drugs. The techniques used should thus enable the various aspects of the cell life to be addressed. Typical examples are the total length of the cell cycle, the length and variability of distinct cell cycle phases, the presence of quiescent or senescent cells, or the presence of death cells.

This review is focused exclusively on the basic techniques of cell cycle analysis in asynchronous cell population. Although techniques based on synchronization protocols have also been developed, we did not include them because their use can result in unwanted perturbation of cell metabolism or they exhibit low efficiency. For example, the use of very common synchronization protocols based on thymidine, mimosine, or aphidicolin may lead to growth imbalance and can also induce imbalance in the expression of cell cycle regulatory proteins such as cyclins B1, A, and E [3]. Less common and emerging techniques of cell cycle analysis are reviewed, e.g., in [4] or [5].

## 2. Growth Models

Presently, several models of population growth are available. In the following subsections, some of the most frequently used are described.

### 2.1. Exponential Growth Model

Cultivated cells without growth limitations such as nutrient depletion, pH limits, senescence induction, or contact inhibition increase their population at a rate proportional to the population size. The growth rate can be expressed by the following equation [6]:(1)$$\frac{\mathrm{dP}}{\mathrm{dt}}=k\times P,$$
where P = P(t) and represents the population as a function of time; k is the growth rate and is equal to the difference of the birth rate and the mortality rate. After integration, the following equation can be obtained [6]:(2)$$P\left(t\right)=P\left(0\right)\times e^{k\times t},$$
where P(t) represents the population size at time t; P(0) is the population size at time 0. The described model is called an exponential growth model. From this equation, the growth rate k can be easily calculated:(3)$$\frac{P\left(t\right)}{P\left(0\right)}=e^{k\times t}$$
(4)$$\ln\left(\frac{P\left(t\right)}{P\left(0\right)}\right)=k\times t$$
(5)$$k=\frac{\ln\left(\frac{P\left(t\right)}{P\left(0\right)}\right)}{t}$$

When the calculated value of k is multiplied by 100, it expresses the percentage increase of the population per time unit [6,7]. An example of an exponential growth curve is shown in Figure 1a.

A similar equation can be deduced from the fact that cell division produces two cells from one. Therefore, any new generation should result in the doubling of the previous cell number. If the mean generation time is G, then the population size after time t can be expressed as:(6)$$P\left(t\right)=P\left(0\right)\times 2^{\frac{t}{G}}$$

However, this equation can be applied only for completely cycling and healthy cell populations. In reality, in cell populations, some cells die and others stop cycling. Therefore, instead of the generation time, the doubling time is used. During the doubling time, the initial number of cells is doubled. If the doubling time is D, then the population size after time t can be expressed as follows:(7)$$P\left(t\right)=P\left(0\right)\times 2^{\frac{t}{D}}$$

The doubling time can then be calculated:(8)$$\frac{P\left(t\right)}{P\left(0\right)}=2^{\frac{t}{D}}$$
(9)$$\log\left(\frac{P\left(t\right)}{P\left(0\right)}\right)=\frac{t}{D}\times \log2$$
(10)$$D=\frac{t\times \log2}{\log\left(\frac{P\left(t\right)}{P\left(0\right)}\right)}$$

The doubling time D is never higher than the generation time G. From Equations (3) and (8), the relation between the growth rate k and the doubling time D can be also expressed:(11)$$e^{k\times t}=2^{\frac{t}{D}}$$
(12)$$\ln(e^{k\times t})=\ln(2^{\frac{t}{D}})$$
(13)$$k\times t=\frac{t}{D}\times \ln2$$

Therefore, the growth rate can be expressed as
(14)$$k=\frac{\ln2}{D},$$
and the doubling time can be expressed as
(15)$$D=\frac{\ln2}{k}$$

### 2.2. Logistic Growth Model

Although the exponential growth model is usually used for the estimation of the cell cycle characteristics of eukaryotic cells, in reality, the environment imposes limitations on the population growth. The exponential phase of growth is short-lived, as exponential growth is only one phase of growth for cell populations. This logistic model postulates that the relative growth rate decreases when the population size P approaches the carrying capacity K of the environment [6]. The logistic growth model was first described by Verhulst in 1838 [6,8,9]. The growth rate is thus expressed by this differential equation [10]:(16)$$\frac{\mathrm{dP}}{\mathrm{dt}}=k\times P\times \left(1-\frac{P}{K}\right)$$

When integrated, the following equation results:(17)$$P\left(t\right)=\frac{K}{1+A\times e^{-k\;x\;t}},\;\mathrm{where}\;A=\frac{K-P\left(0\right)}{P\left(0\right)}$$

An example of the logistic growth curve is shown in Figure 1b.

### 2.3. Monod Kinetics Model

Monod kinetics models belong to the growth models used mainly in studies of microorganisms. They relate the population growth rate to the concentration of the limiting resources [6,11]. This growth model can be expressed by the following equation [6]:(18)$$k=k_{\max}\times \frac{S}{K_{S}+S},$$
where k_max_ is the maximum growth rate; S is the concentration of the limiting resource necessary for growth, and K_s_ is the concentration of the limiting resource where the growth rate is half of the maximum. In this model, it has to be taken into account that sometimes more than one limiting resource exists. In this case, multiple equations of (18) can be multiplied together to describe the population growth kinetics [6].

### 2.4. Allee Effect Model

The Allee effect model emanates from the logistic growth model [6,12,13,14]. In this case, the size of the population affects the individual growth. This model is mainly applied in ecology; however, it has also been used in modeling of cancer cell populations [6,15].

## 3. Cell Cycle and Markers of the Cell Cycle Phases

The cell cycle is composed of four distinct phases. They are the G1 (gap 1), S (DNA synthesis), G2 (gap 2), and M (mitosis) phases (Figure 2). The progression of cells through the cell cycle is controlled by highly orchestrated steps reacting to intracellular and extracellular signals. An important role in these steps is played by cyclin-dependent kinases (CDKs; Figure 2) regulated by their interactions with cyclins and CDK inhibitors (CKIs; for a review see, e.g., [16]). The formation of cyclin/CDKs complexes controls the cell cycle progression by the phosphorylation of the targets, such as tumor suppressor protein retinoblastoma (Rb). The activation of cyclins/CDKs is induced by mitogenic signals and inhibited by the activation of the cell cycle checkpoints in response to DNA damage [16,17]. In human cells, there are supposed to be around 20 various CDKs and 29 various cyclins [16,18]. During the cell cycle, several important control points (checkpoints) are present (Figure 2). In the early G1 phase, until cells reach the restriction point, the growth factors are required for their transition through the G1 phase leading to the S phase [19]. When cells pass through the restriction point, no growth factors are necessary for transition to the S phase [19,20]. Another important checkpoint is the G1/S checkpoint that controls the DNA integrity and stops DNA synthesis when cells suffer, for example, from extracellular stresses. The G2/M checkpoint prevents cells with damaged DNA from entering mitosis and allows for DNA repair [21].

It is well known that mammalian cells exit the cell cycle in response to environmental changes such as depletion of nutrition or growth factors, changes in cell adhesion, or increased cell density during the early G1 phase. This state is called the quiescent (G0) phase. Quiescence is an important feature of stem cells. Moreover, many types of differentiated cells are in the G0 phase in vivo [22].

In addition, some cells escape the cell cycle during senescence. In contrast to cellular quiescence, it results in cell arrest in the G1 and/or G2 phases [23]. Senescence can be triggered by developmental signals or by different types of stress [23]. For example, cells can undergo senescence in response to various intrinsic and extrinsic stimuli, such as progressive telomere shortening, changes in the telomeric structure, mitogenic signals, oncogenic activation, radiation, oxidative and genotoxic stress, epigenetic changes, chromatin disorganization, perturbed proteostasis, mitochondrial dysfunction, inflammation and/or tissue damage signals, irradiation or chemotherapeutic agents, or nutrient deprivation [23,24,25,26,27,28,29].

Various approaches are used to identify the distinct cell cycle phases. The approach that is likely the most frequent is based on the analysis of DNA content. The cells in G1 and G0 have half the content of DNA as compared to G2 and M cells. During determination of the amount of DNA, DNA is stained by fluorescent dyes, and the fluorescent signal is analyzed. The histogram of DNA content thus enables the estimation of the number of cells in G1/G0, S, and G2/M phases. The most frequently used protocols for DNA staining of fixed cells are based on the use of fluorescent, stoichiometrically DNA binding dyes such as propidium iodide (PI, Figure 3), 4,6-diamidino-2-phenylindole (DAPI), Hoechst 33258, or Hoechst 33342 [30]. For the staining of the living cells, Hoechst 33342 is commonly used [31]. As PI also stains RNA, the protocols based on PI also include the step with the treatment of cells with RNase [32]. As the cells with the same DNA content can differently bind fluorescent dyes to their DNA [33], DNA content does not allow easy discrimination between the G1/G0, S, and G2/M phases. Thus, mathematical approaches were developed for the estimation of fractions of cells in the G1/G0 phases, S phase, and G2/M phases [34,35,36,37,38].

For the clear distinguishing of cells in the S phase, methods based on the incorporation of modified nucleosides followed by their staining are commonly used. The most frequently used precursors are 5-bromo-2′-deoxyuridine (BrdU) and 5-ethynyl-2′-deoxyuridine (EdU) [39,40]. The detection of cells in the S phase enables the distinguishing of the G1/G0 phase from the G2/M phase without the need to use the mathematical approaches mentioned above.

For the distinguishing of mitotic cells, the detection of histone H3 phosphorylated on Ser-10 is usually applied [41].

The determination of G0 cells can be achieved by the application of pyronin Y for staining of RNA [42,43]. It relies on the assumption that cells in the G0 phase have a lower level of RNA compared to the G1-phase cells, and therefore it allows for the distinguishing of G0 cells. Another option for G0 phase identification is Ki-67 antibody staining [44]. Initial studies on the Ki-67 protein indicated that the Ki-67 protein is present during every phase of the cell cycle in asynchronously cycling cells and absent in non-dividing cells [45]. Recent studies on the Ki-67 protein have indicated that this protein undergoes proteasome-mediated degradation during the G1 phase and upon cell cycle exit. On the other hand, the depletion of the Cdh1 protein, which is an activator of the Anaphase Promoting Complex, stabilizes the Ki-67 protein [46,47,48]. The level of Ki-67 protein during G0 and G1 phases in individual cells is highly heterogeneous. It depends on the time that an individual cell has spent in the G0 phase. Therefore, the Ki-67 protein is degraded continuously during the G1 and G0 phases and thus is a graded rather than a binary marker, both for cell cycle progression and time since entry into quiescence [46].

Markers for the clear discrimination of senescent cells are not available (see note 1).

Besides these markers, the in vivo reporters of cell cycle phases can be used. A very common approach is based on the use of a fluorescent protein-based indicator system to monitor the cell cycle status, namely, the fluorescent ubiquitination-based cell cycle indicator (Fucci) [49]. In this system, G1 phase-specific proteolysis of the protein geminin and S/G2/M phase-specific proteolysis of the Cdt1 protein are monitored using two types of probes consisting of the fusion proteins between the degrons of geminin and of Cdt1 to fluorescent proteins. The Fucci system differentially labels the cells in the G1 phase and those in the S/G2/M phase, effectively visualizing the G1-S and M-G1 transitions. However, Fucci cannot be used to distinguish the cells in the G0 phase from those in the G1 phase, since Cdt1 is expressed in both phases [22,49]. For identification of G0 cells, the fusion protein consisting of mVenus, and a defective mutant of CDK inhibitor, p27 (p27K−) was prepared and used. The Fucci system was later improved by the addition of two fluorescently labeled proteins: stem-loop binding protein (SLBP), an RNA-binding protein that is degraded following the S phase [50], and histone H1.0, which is used for the identification of chromatin condensation during mitosis.

## 4. Determination of Fractions of Cells in Distinct Phases of the Cell Cycle

The most common approach that enables the monitoring of cell cycle progression is based on the static observation of the situation in the cell population at a specific time. If DNA and/or other markers of specific cell cycle phases are used, mutual information is provided about the fractions of cells in a specific phase of the cell cycle.

Data acquisition and analysis are usually performed by flow or image cytometry. Although common flow cytometry is very fast, it requires samples in the form of cell suspension. Moreover, it does not allow the signal distribution to be taken into account as image cytometry does. On the other hand, data processing and analysis using image cytometry are time-consuming and require the use of appropriate software applications (see note 2).

The routine flow cytometry analysis based on DNA content alone is commonly called univariate analysis. This type of analysis is very common; however, to gain more accurate separation of particular cell cycle phases, the above-mentioned markers of the cell cycle phases are used as well [40].

## 5. From the Fractions to the Cell Cycle Phase Length

It is evident that the fraction of cells in asynchronous cell population in different cell cycle phases are related to the time spent in these phases. For example, much fewer cells are observed in the M phase than in the G1 phase, as the M phase duration is much shorter than that of the G1 phase. For non-synchronous cycles without division (no offspring), the time spent in any phase is equal to the multiple of the fraction of cells in this phase and the length of the cycle.

In contrast to non-dividing systems, a system with offspring exhibits a different relation between the time spent in the particular cell cycle phase and the generation time. Although intuitively it seems that the number of cells at the beginning of the G1 phase (e.g., in the first minute of the G1 phase) is the same as in the last minute of the M phase, this is not true. The frequency distribution is skewed, and there are always more young cells (early in the cycle) than older cells (late in the cell cycle) [51]. For cell cycles with two offspring, the relation between the observed portion of cells in the first stage (P_G1_) of the cell cycle, generation time (G), and time (t) spent in this phase is expressed by the following equation [52]:(19)$$t=-G\times \frac{\ln\left(1-\frac{P_{G1}}{2}\right)}{\ln2}$$

For an unrealistic but illustrative example of a cell population with around one third of its cells in the G1 phase, one third in the S phase, one third of in the G2/M phase, and a generation time of 24 h, the length of the particular cycle phases can be calculated as follows:(20)$$G1\;\mathrm{length}=-24\times \frac{\ln\left(1-\frac{1}{6}\right)}{\ln2}=6.31\;h$$
(21)$$G1\;\mathrm{length}+S\;\mathrm{length}=-24\times \frac{\ln\left(1-\frac{2}{6}\right)}{\ln2}=14.04\;h$$
(22)S length=14.04−6.31=7.73 h
(23)G2/M length=24−14.04=9.96 h

For the deduction of the length of the cell cycle phases close to the cycle end, we can use a different equation. In this case, the portion of cells in the last stage of the cell cycle is used [53]:(24)$$t=G\times \frac{\ln\left(P_{M}+1\right)}{\ln2}$$

It is very helpful if the mitotic index (portion of cells in the M phase) is known, as the length of the mitotic phase can be estimated quickly by substituting the mitotic index for P_M_.

For cell cycles with an arbitrary number of proliferative offspring (such as multiple fission, or a chance of terminal differentiation), the following equation can be used [52]:(25)$$t=G\times \frac{\ln\left(q\right)-\ln\left(q-\left(q-1\right)\right)\times P_{G1}}{\ln q},$$
where q is the number of proliferative offspring per division. This equation uses the portion of cells from the first stage of the cell cycle (G1). The last equation is useful, for example, in situations when the stem cells generate one non-proliferative daughter. For instance, if the proliferating stem cell has a 70% chance of generating one non-proliferative daughter, the value of q is equal to 1.3 (2–0.7).

The type of the above-presented mathematical analyses of asynchronous cell populations can be termed ergodic analysis [52], which is based on two assumptions:The weak ergodic assumption—If the distribution of cells among different states does not change over time, then the proportion observed in any state is proportional to the time each cell spends, on average, in that state.Strong ergodic assumption—If all cells are going through an identical cycle of events, then the proportion of cells in any cycle stage observed in the population at a single time point is the same as the proportion of time spent in that cycle stage as a single cell progresses through the cycle [52].

Ergodic analysis is a very powerful tool in the case in which the assumption is met. Ergodic analysis can also be used for the determination of the length of various cellular processes if the marker and the position of the process in the cell cycle is known.

On the other hand, any ergodic analysis should be accompanied by an analysis of these assumptions. For example, in the case of ergodic analysis of the length of G1, S, G2, and M phases of non-cancer cells, the analysis of senescence and induction of transit from the G1 to the G0 phase should be performed simultaneously.

If the values of absolute lengths of the cell cycle phases are required, the knowledge of the generation time or the length of at least one phase is necessary. The value of the generation time can be calculated from the doubling time. From the expression of the relation between the doubling time and the growth rate (14) and (15), it is, however, evident that the calculation of the generation time is possible only if the mortality rate and the fraction of non-cycling cells is known, as the growth rate k is equal to the difference of the birth rate and the mortality rate.

## 6. Approaches Based on Kinetic Analysis

Typically, kinetic approaches based on the labeling of DNA by nucleoside analogs or on time lapse microscopy are used for the evaluation of the cell cycle. An overview of the methods is summarized in Table 1.

### 6.1. Approaches Based on DNA Labeling

#### 6.1.1. Marker Nucleosides

The first labeled nucleosides used radioisotopes for their detection. In 1957, Taylor and colleagues developed a method where they used H3-thymidine for the labeling of replicated DNA in the bean root and autoradiography for its detection [60,61]. This technique was gradually forced out by new approaches. Perhaps the most important limitations of the autoradiography approach are that it is time consuming and exhibits low resolution [61].

The halogen derivatives of thymidine were the first widely used alternative to the isotopically labeled nucleosides. These thymidine analogs are modified in the 5 position of the thymine ring by halogen atoms (bromine, chlorine, and iodine). Although BrdU is the most frequently used [60], chlorine (CldU) [62] and iodine analogs (IdU) [63] can be used as well. Although CldU and IdU are not as frequently used as BrdU, they allow the performance of pulse–chase–pulse experiments with two different labeling pulses [64]. In this case, the double labeling of replicating DNA is based on two different clones of primary antibodies raised against BrdU. One of these antibodies reacts with CldU but not with IdU. The second antibody clone reacts both with CldU and IdU. After washing in a buffer with a high salt concentration, the second antibody linked to CldU-labeled DNA is washed out [64].

As BrdU, CldU, and IdU are inaccessible in double-stranded DNA for reactions with antibodies, it is necessary to use special treatments to make them accessible for such reactions [60,65,66,67,68,69,70,71,72,73,74,75]. Due to the necessity of disrupting the structure of double-stranded DNA in the case of halogen derivatives of thymidine, other methods of detection of replicating DNA were investigated. In 2008, Salic and Mitchinson published an alternative approach for the labeling of DNA replication. They used another modified analog of thymidine, namely, EdU. EdU has alkyne instead of halogen, which can react with the fluorescent azides (such as Alexa-azide stains) in what is termed the click reaction [60,76]. Although the click reaction is a very fast approach without the necessity of the additional exposing of incorporated EdU in DNA, the toxicity of EdU is relatively high (see, e.g., [77,78,79,80]). EdU toxicity is usually accompanied by the deformation of the cell cycle, the slowdown of the S phase, and the induction of interstrand crosslinks. Taken together, although EdU is extremely useful for short-term experiments, it is not suitable for long-term studies [78]. In this respect, BrdU can induce metabolic changes as well; however, its toxicity is much lower than that of EdU. Although prolonged incubation with BrdU can result in senescence induction, the effective concentration for the induction of senescence is usually much higher than the concentration used for DNA labeling [81].

#### 6.1.2. Mitotic Window Approach

The mitotic window approach is based on the short incubation of cells with the labeled nucleosides and is used for looking for labeled cells through the mitotic window. Originally, this method was based on radioactive isotopes [82,83]. However, BrdU or EdU are currently more convenient markers. During the procedure, cells in the S phase are labeled using short pulse with the labeled nucleosides, chased for a gradually increased period, and then fixed. In the next step, the mitotic cells are analyzed. The first labeling of mitotic cells occurs at the time when the first labeled cells reach the mitotic phase. The number of labeled mitotic cells grows quickly to 100%. If most of the labeled cells traverse the M phase, the number of labeled cells declines quickly. The time distance of the midpoint of the rise from the end of labeling is used for estimation of the G2 phase. The time distance between the midpoint of the rise and the midpoint of the decline is used for estimation of the S-phase length [54].

The precision of the mitotic window method depends mainly on the length of periods between the sample collections. Its advantage is insensitivity to the presence of senescent or quiescent cells.

#### 6.1.3. Continuous Labeling Method

In this approach, cells are incubated with a labeled nucleoside for gradually increased times, and a portion of the labeled cells is evaluated. The rise of the signal is stopped if all cells are labeled. As all cells in the S phase are labeled by a pulse at its beginning, the maximum is reached after a time that is equal to the G2 + M + G1 phases. If less than 100% of cells are labeled, this indicates that the cell population contains non-cycling cells as well [54].

As in the case of the mitotic window approach, a shorter period between samples results in a higher precision of obtained results. Senescent or quiescent cells cause decrease of the maximum labeled cells. On the other hand, it should not exhibit an effect on the estimation of the G2 + M + G1 phase length [54].

A different approach for the analysis of continuously labeled cells was applied by Pereira and colleagues [55]. In this study, cells were labeled with EdU for gradually increased times, and the signal intensity was then analyzed. Maximal EdU-coupled fluorescence intensity was reached if pulsing times matched the length of the S phase. However, this approach can apparently result in the overestimation of the S-phase length if a variability in S-phase length exists. Moreover, the eventual effect of toxicity of EdU (e.g., [78]) should be evaluated in control experiments.

#### 6.1.4. Double Labeling Approach

This method is based on the use of two labeled nucleosides. Several arrangements of this experiment are possible to use.

According to Hwang and colleagues, the sequential labeling of cells using EdU followed by staining with BrdU enables the estimation of the length of the cell cycle [56]. In this case, the EdU pulse is followed by the BrdU pulse after a known time interval. During the pulse with EdU, some cells leave the S phase. This portion is identified by the omission of the BrdU signal. It is used for estimation of the cell cycle length using the following equation:(26)$$G=\frac{I}{f},$$
where G is the cell cycle length; I is the time period between EdU and BrdU labeling, and f is the fraction of cells labeled exclusively by EdU. As f is the fraction related to all cells, the method is sensitive to the presence of quiescent or senescent cells and to cell mortality.

A similar experimental arrangement was used by Martynoga and colleagues [57]. They exposed cells first to IdU and later also to BrdU. The fraction of cells labeled by the first pulse exclusively and all cells labeled by the second pulse served for the estimation of the S-phase length. In this case, the S-phase length calculation was based on the assumption that the ratio between these two cell fractions is equal to the ratio between the length of the time interval between two labeling pulses and the length of the S phase [57]. S-phase length was calculated by the following equation:(27)$$S=I\times \frac{\mathrm{all}\;\mathrm{BrdU}-\mathrm{labeled}\;\mathrm{cells}}{\mathrm{exclusively}\;\mathrm{IdU}-\mathrm{labeled}\;\mathrm{cells}},$$
where S is the length of the S phase, and I is the time interval between the two labeling pulses.

The generation time was estimated using the equation:(28)$$G=S\times \frac{\mathrm{all}\;\mathrm{cells}}{\mathrm{all}\;\mathrm{BrdU}-\mathrm{labeled}\;\mathrm{cells}},$$
where G is the generation time, and S is the length of the S phase [57].

Another experimental arrangement was developed to estimate the length of the S phase as well [58]. In this case, cells were briefly labeled with EdU and then using BrdU or pulses were mutually separated by the gradually increased chase period. As some cells labeled with EdU left the S phase during this chase period, the number of cells labeled by both pulses decreased with the prolongation of the chase length. The length of the shortest chase providing zero double-labeled cells allowed for the estimation of the length of the S phase. To eliminate the possibility that EdU-labeled cells would move to the second pulse, nocodazole was applied during the chase period [58]. The procedure tends to provide a higher length of the S phase than is the real mean value if variability in S-phase length exists. On the other hand, it should not be sensitive to the presence of quiescent and senescent cells.

### 6.2. Methods Based on Time Lapse Microscopy

Time lapse microscopy approaches are based on microscopes equipped with chambers to maintain a constant temperature and atmosphere. Images of cells from areas of interest are acquired on a regular time interval (e.g., every 10 min). For the discrimination of cell cycle phases, cells expressing fluorescently labeled reporters are commonly used. For example, Chao and colleagues used fluorescently labeled PCNA (Figure 4) [59]. PCNA (proliferating cell nuclear antigen) is a protein involved mainly in DNA replication, but its functions are also associated with chromatin remodeling, DNA repair, sister-chromatid cohesion, and cell cycle control [84,85]. In the S phase, PCNA is loaded at DNA replication forks forming clear punctuate pattern (see, e.g., [59,71,86]). The transition of the punctuate to the diffuse pattern (S/G2 phases) and from the diffuse to the punctuate pattern (G1/S phases) can be readily detected between consecutive frames of the time-lapse imaging [59].

Chao and colleagues used, in the above-mentioned study, pLenti-PGK-Puro-TK-NLS-mCherry-PCNA as a marker of cells in the S phase [59]. The transition from the diffuse to the punctate pattern (G1/S phase) and from the punctate back to the diffuse pattern (S/G2 phase) was used to discriminate between G1, S, and G2 phases. The G2/M transition was identified by the nuclear envelope breakdown [59]. Other examples of the typical reporter system are studies using the FUCCI vector system [49]. Approaches based on time lapse microscopy can address the situation in individual cells; therefore, they can provide interesting data about the variability of the length of the cell cycle and cell cycle phases inside the cell population. In this respect, the stretched cell cycle model for proliferating lymphocytes was suggested using time lapse microscopy [87]. According to this model, all parts of the cell cycle of proliferating lymphocytes are proportional to the total division time.

Another example of the importance of time lapse microscopy is data showing that at least in some cell lines, each phase duration follows an Erlang distribution and is statistically independent of other phases [59]. Lastly, time lapse microscopy was also used to address a very important aspect of the cell cycle, namely, cell death kinetics [88].

## 7. Conclusions and Future Perspectives

It is evident that various approaches can be used to characterize cell cycle progression. While approaches based on the static observations of DNA content and markers of distinct phases are fast and extremely useful for routine analysis of the cell cycle, they provide only relative data about the length of distinct cell cycle phases. Even if these studies are accompanied by the analysis of doubling time, the accurate estimation of the length of cell cycle phases is frequently very difficult, as such estimation requires knowledge of the mortality rate and the fraction of non-cycling cells as well. Despite the existence of various markers of cell death, such as propidium iodide or 7-amino-actinomycin D [89,90], these approaches can only serve to estimate the relative changes of the mortality rate, e.g., between the control and treated cells. Moreover, although several markers of quiescent cells are available, unambiguous markers of senescent cells are not yet known.

This gap can be overcome by kinetic approaches based on DNA labeling, as some of them are insensitive to the presence of quiescent and senescent cells. These approaches can also provide a rough estimation of variability of the cell cycle phases. On the other hand, the toxicity of the marker nucleosides must be taken into account.

Although time lapse studies provide very accurate data about the cell populations of observed cells, including data about variability of different phases, such studies are lengthy and expensive, as automatic microscope stations, chambers with a controlled environment, software solutions, and corresponding reporting systems are necessary. In addition, phototoxicity can affect cell cycle progression, as damage to cellular macromolecules upon excitation light illumination can impair sample physiology and even lead to sample death [91].

Apparently, we can expect further progress in the automatization of sample preparation and processing, further automatization of microscope stations, progress in machine learning, solutions resulting in the decrease of phototoxicity, and also in the development of new markers of distinct cell cycle phases. All these advances are necessary for the development of highly efficient solutions for reliable analysis of the cell cycle by time lapse microscopy. The requirement for efficient software solutions is evident from the fact that even with a moderate duration of a few days, time lapse movie data output can range from hundreds of gigabytes up to terabytes or more with recently developed hardware in wide-field and light-sheet microscopy. The exploration of these multidimensional, information-rich data thus requires the use of efficient automatic cell segmentation, tracking, and phenotype quantification algorithms [92].

Some of these advances can also contribute to the improvement of flow or image cytometry. For example, new markers can help to discern senescent cells or simplify the recognition of cells in the S phase. Similarly, any improvement in the identification and segmentation algorithm will also impact results from image cytometry, as in contrast to flow cytometry, these algorithms are crucial for the output data quality. In this respect, progress in imaging flow cytometry, which combines the possibility of fast identification of individual cells with the analysis of cell morphology, can substantially accelerate any analysis requiring large numbers of cells, as the omission of the segmentation is crucial for the acceleration of data processing [93]. Similarly, progress in mass flow cytometry substantially expands the possibility of cell cycle characterization, as it provides the possibility to use a high number of simultaneously labeled antibodies against cell cycle markers [94].

## 8. Notes

Below are listed some of the commonly used markers of senescent cells. However, none of the senescence-associated markers are exclusive and typical for senescent cells; therefore, usually at least two different markers are detected
(a)Senescent cells are much bigger than non-senescent counterparts [95,96,97]; however, exceptions exist [98].(b)Senescent cells contain increased numbers of lysosomes [99,100]. This finding resulted in the development of a technique based on the detection of activity of the senescence-associated (SA) β-galactosidase at pH 6 [100]. Unfortunately, a significant delay between the senescence entry and SA β-galactosidase-derived staining was also previously reported [101]. Moreover, intense SA β-galactosidase labeling could also reflect an alteration in lysosomal number or activity in non-proliferating cells [102] or in terminally differentiated cells such as neurons [103]. Furthermore, endogenous SA β-galactosidase activity was found also in confluent non-transformed fibroblast culture [104].(c)Senescent cells contain γH2AX foci [105]. However, these foci can also be observed in non-senescent cells [106].(d)Senescent cells contain senescence-associated heterochromatin foci [107]. On the other hand, foci formation is cell-line dependent and is mostly associated with oncogene-induced senescence [24,108,109].Below is a list of freeware software that can be used for image analysis
(a)ImageJ/FIJI—ImageJ is widely used software for processing and analyzing scientific images; FIJI is a “batteries-included” distribution of ImageJ and ImageJ2, which includes many useful plugins contributed by the community. It can be downloaded from https://imagej.net/downloads (accessed on 9 January 2023) [110].(b)CellProfiler^TM^—CellProfiler is software allowing for automatic processing and analysis of sets of images according to the instructions in the form of user-defined pipelines. It can be downloaded from https://cellprofiler.org (accessed on 9 January 2023) [111].(c)CellProfiler Analyst^TM^—CellProfiler Analyst allows data to be explored from CellProfiler through interactive visualizations that link to images, classifying complex or subtle phenotypes using machine learning. It can be downloaded from https://cellprofileranalyst.org (accessed on 9 January 2023) [112].(d)Ilastik—ilastik is software that applies machine learning algorithms to easily segment, classify, track, and count the cells or other experimental data. It can be downloaded from https://www.ilastik.org/download.html (accessed on 9 January 2023) [113].

## Figures and Tables

**Figure 1 ijms-24-03674-f001:**
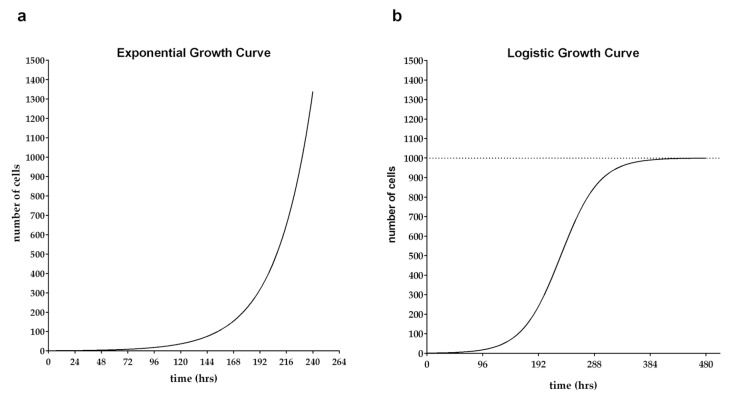
Models of the growth curves: (**a**) exponential growth curve; (**b**) logistic growth curve. (**a**) The exponential growth curve was modeled in GraphPad Prism 6 software (Dotmatics, Boston, MA, USA) using Plot a function analysis using the Exponential growth equation with the following parameters: P(0) = 1; k = 0.03 (k was derived from Equation (14) with the doubling time D equal to 24 h). (**b**) The logistic growth curve was modeled in GraphPad Prism 6 software (Dotmatics, Boston, MA, USA) using Plot a function analysis using the Logistic growth equation with the following parameters: P(0) = 1; k = 0.03 (k was derived from Equation (14) with the doubling time D equal to 24 h; K = 1000, where K is the maximum cell population).

**Figure 2 ijms-24-03674-f002:**
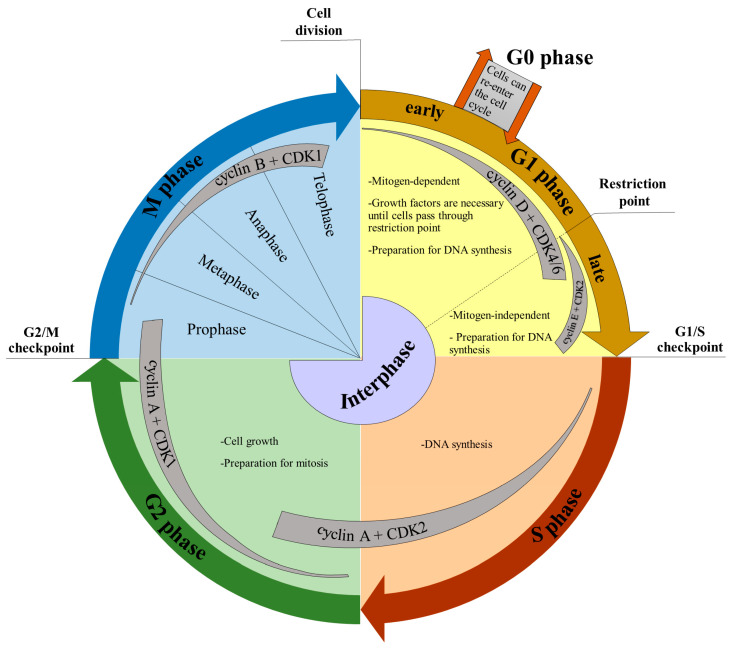
Schema of the cell cycle.

**Figure 3 ijms-24-03674-f003:**
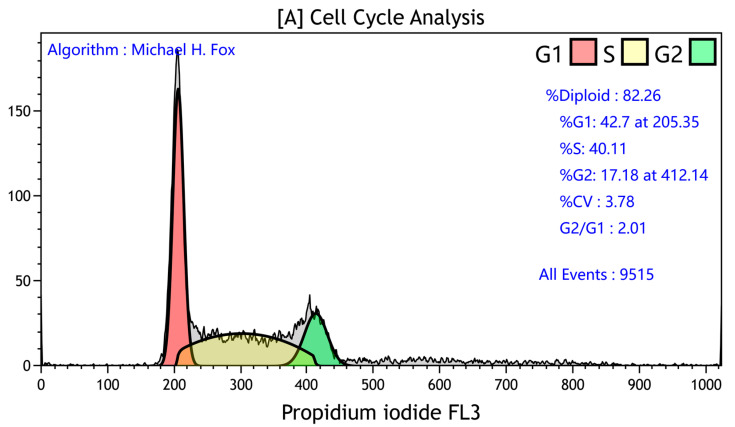
An example of cell cycle analysis using PI. The percentage of cells in distinct cell cycle phases calculated using Kaluza Analysis Software 2.2 (Beckman Coulter, Brea, CA, USA) based on the Michael H. Fox algorithm.

**Figure 4 ijms-24-03674-f004:**
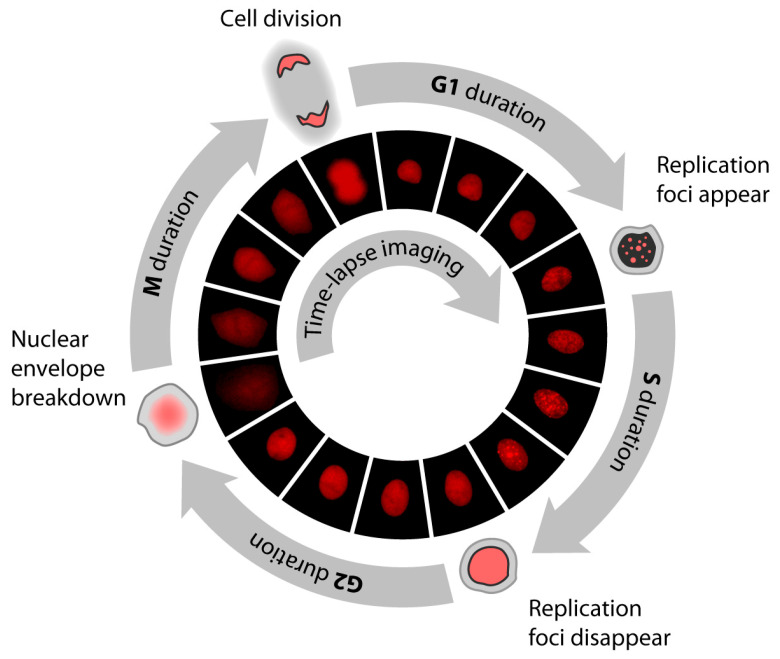
Time lapse analysis of the cell cycle using fluorescently labeled PCNA. Scheme of the cell cycle composed of G1, S, G2, and M phases (not to scale). Duration of particular cell cycle phases were quantified using time lapse fluorescence microscopy by a PCNA-mCherry reporter to identify four discrete events during the lifetime of an individual cell. Fluorescence images were acquired using a Nikon Ti Eclipse inverted microscope with a Nikon Plan Apochromat Lambda 40× objective with a numerical aperture of 0.95 using an Andor Zyla 4.2 sCMOS detector. In addition, the Nikon Perfect Focus System (PFS) was employed in order to maintain focus of live cells throughout the entire acquisition period. The microscope was surrounded by a custom enclosure (Okolabs) in order to maintain a constant temperature (37 °C) and atmosphere (5% CO_2_). NIS-Elements AR software was used for image acquisition and analysis. Images were acquired every 10 min. Adapted from [59].

**Table 1 ijms-24-03674-t001:** Overview of the approaches for cell cycle evaluation based on kinetic analysis.

Method (Suitable for Estimation)	Principle	Common Markers	Notes	Ref.
Mitotic window (G2 and S phases)	Analysis of the signal from the replicated chromatin in mitotic cells.	EdU, BrdU	Time consuming. Variability of G2 and S phases can be addressed.	[54]
Continuous labeling based on cell number of the labeled cells (sum of G2, M, and G1 phases)	Analysis of the number of labeled cells after continuous labeling of replicated chromatin.	EdU, BrdU	Time consuming. Long incubation time with the potentially toxic marker nucleoside. BrdU is less toxic than EdU.	[54]
Continuous labeling based on the signal intensity of the labeled cells (S phase)	Analysis of the signal intensity after continuous labeling of replicated chromatin.	EdU	The toxicity of EdU should be tested before its use.	[55]
Double labeling (S phase, generation time)	Analysis of the number of labeled cells after labeling of replicated chromatin by two consecutive pulses with the different replication marker nucleosides.	EdU, BrdU, IdU	Fast. The sensitivity to the presence of G1 and senescence cells.	[56,57]
Double labelling with chase period (S phase)	Analysis of the number of labeled cells after labeling of replicated chromatin by two consecutive pulses with different replication marker nucleosides separated by different chase periods.	EdU, BrdU	Nocodazole treatment is required.	[58]
Time lapse microscopy (all phases)	Analysis of living cells expressing fluorescently labeled reporters at regular time interval	Fluorescently labeled PCNA, Ctd1, Geminin	Time consuming. Special equipment is necessary. Phototoxicity can affect the results. Variability of different cell cycle stages can be addressed. Mortality rate can be measured.	[49,59]

## Data Availability

No new data were created or analyzed in this study. Data sharing is not applicable to this article.

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
