# Peer review of "Basic Methods of Cell Cycle Analysis"

_ijms, 2023, doi:10.3390/ijms24043674_

Round 1

Reviewer 1 Report

In this review, Ligasova and collaborators describe several methods to analyze cell cycle phases.

The paper is interesting and deserves publication after few modifications.

Specific points:

-       In figure 2, checkpoints are indicated but never cited in the text. The authors must add a sentence about checkpoint in paragraph 3 or remove the checkpoint from figure 2.

-       Some typos must be corrected (es: at the end of page 7 “S legth” must be “S length”)

-       Paragraph 6.2 the authors must explain what PCNA is and which fluorophor has been used to mark it.

-       In figure 4 the quality of the micro photographs is not excellent. If possible the authors must change them and they must indicate which microscope, objective and software of acquisition have been used. The authors must add which type of cells have been used.

-       the authors must expand the last paragraph of chapter 7 about improvement of flow cytometry

Author Response

Thank you for your comments.

1, In figure 2, checkpoints are indicated but never cited in the text. The authors must add a sentence about checkpoint in paragraph 3 or remove the checkpoint from figure 2.

We added short paragraph concerning the checkpoints (page 4, lines 121-128).

2, Some typos must be corrected (es: at the end of page 7 “S legth” must be “S length”)

We corrected the typos.

3, Paragraph 6.2 the authors must explain what PCNA is and which fluorophor has been used to mark it.

We added the requested information about PCNA and fluorophore (page 12, lines 385-393).

4, In figure 4 the quality of the micro photographs is not excellent. If possible the authors must change them and they must indicate which microscope, objective and software of acquisition have been used. The authors must add which type of cells have been used.

Figure 4 is a reused Figure from the original article of Chao and colleagues (https://www.embopress.org/doi/full/10.15252/msb.20188604). We added to the legend of the Figure 4 the information about the microscope, objective, and software of acquisition (page 13, lines 408-414). As authors did not specify which cell line is on their Figure and they used three different cells lines in their experiments, we are not able to specify the concrete cell line on the Figure.

5, the authors must expand the last paragraph of chapter 7 about improvement of flow cytometry

We expanded the last paragraph of Chapter 7 (pages 13-14, lines 457-463).

Reviewer 2 Report

In this paper, the authors describe the basic methods used in determining cell cycle phases with reference to their efficiency and reproducibility. There are already journal articles on this topic, and yet, it can be said that this review in one place included an overview of the methods that are most often used in research today. The presented results were well interpreted and appropriate conclusions were drawn from them. The study is well-designed and the idea is good. In the review itself, the authors also used their previous results, thus supporting this area of research and thus the motive to write about this topic. The English language is acceptable and understandable.

Author Response

Thank you for your comments.

Reviewer 3 Report

The manuscript “Basic Methods of Cell Cycle Analysis” by Ligasova et al. presents the basic methods used to monitor the cell cycle phases. First, the authors explain some models of the population growth, following by the introduction of the basic principle regulating cell cycle and discussion of various approaches to identify and distinguish cells in particular cell cycle phases. Overall, the manuscript is well written, has a logical concept and is easy to follow.

Minor Comments:

It would be useful for readers, if the methods communicated in sections "6. Approaches based on kinetic analysis” will be summarized in the form of Table.

Author Response

Thank you for your comments. 

1, It would be useful for readers, if the methods communicated in sections "6. Approaches based on kinetic analysis” will be summarized in the form of Table.

We added the Table 1 with the overview of the approaches based on the kinetic analysis (page 9).